# Attempts to Create Transgenic Mice Carrying the Q3924E Mutation in RyR2 Ca^2+^ Binding Site

**DOI:** 10.3390/cells13242051

**Published:** 2024-12-12

**Authors:** Xiao-hua Zhang, Fu-lei Tang, Allison M. Trouten, Martin Morad

**Affiliations:** 1Cardiac Signaling Center, University of South Carolina, Medical University of South Carolina and Clemson University, Charleston, SC 29425, USA; zhaxi@musc.edu; 2Department of Comparative Medicine, Medical University of South Carolina, Charleston, SC 29425, USA; fulei@musc.edu; 3Department of Regenerative Medicine and Cell Biology, Medical University of South Carolina, Charleston, SC 29425, USA; trouten@musc.edu

**Keywords:** RyR2 Q3924E mouse, sarcoplasmic reticulum, sudden death, hiPSC-CMs

## Abstract

Over 200 point mutations in the ryanodine receptor (RyR2) of the cardiac sarcoplasmic reticulum (SR) are known to be associated with cardiac arrhythmia. We have already reported on the calcium signaling phenotype of a point mutation in RyR2 Ca^2+^ binding site Q3925E expressed in human stem-cell-derived cardiomyocytes (hiPSC-CMs) that was found to be lethal in a 9-year-old girl. CRISPR/Cas9-gene-edited mutant cardiomyocytes carrying the RyR2-Q3925E mutation exhibited a loss of calcium-induced calcium release (CICR) and caffeine-triggered calcium release but continued to beat arrhythmically without generating significant SR Ca^2+^ release, consistent with a remodeling of the calcium signaling pathway. An RNAseq heat map confirmed significant changes in calcium-associated genes, supporting the possibility of remodeling. To determine the in situ cardiac phenotype in an animal model of this mutation, we generated a knock-in mouse model of RyR2-Q3924E+/− using the CRISPR/Cas9 technique. We obtained three homozygous and one chimera mice, but they all died before reaching 3 weeks of age, preventing the establishment of germline mutation transmission in their offspring. A histo-pathological analysis of the heart showed significant cardiac hypertrophy, suggesting the Q3924E-RyR2 mutation was lethal to the mice.

## 1. Introduction

Ryanodine receptor type 2 (RyR2) is the major Ca^2+^ release channel of the cardiac sarcoplasmic reticulum (SR) that controls cardiac excitation–contraction coupling (ECC) and myocardial contraction [1,2,3]. Over 200 missense mutations of RyR2 have been reported to induce severe arrhythmia, unexplained sudden cardiac death and heart failure [4,5,6,7]. Three different platforms, including transgenic mice, the recombinant expression of mutant RyR2 in HEK cell lines and stem cell-derived human cardiomyocytes, have been used in previous studies [8,9,10,11]. Transgenic mouse models expressing point mutations or the recombinant expression of mutant RyR2 in HEK cells have yielded significant insights into CPVT1 pathology. The mouse heart model suffers from great differences in heart size and ion channel expression compared to the human heart [12,13,14]. The recombinant RyR2 mutant expression platforms, while providing critical molecular data on the function of RyR2, lack the multiplicity of the calcium signaling pathways of mammalian hearts. This model, therefore, fails to provide for the possible functional mutation-induced remodeling of cardiac signaling pathways. The hiPSC-CM platform, in which the mutations are introduced using gene editing via CRISPR/Cas9 [15,16,17], allows us to examine the mutation effects in cardiomyocytes with a human genetic background, and these cells express the multiplicity of cardiac calcium signaling pathways. Thus, hiPSC-CMs represent an ideal model were it not for the subcellular structural immaturity of these cells, which lack t-tubules and associated dyadic junctions.

In a previous article, we reported on two RyR2 Ca^2+^ binding site mutations, Q3925E and E3848A, created by GRISPR/Cas9 gene editing in human-induced pluripotent stem-cell-derived cardiomyocytes (hiPSC-CMs) [17]. In single isolated cardiomyocytes, these mutations in the RyR2 Ca^2+^ binding site induced marked Ca^2+^ signaling deficiencies and CICR remodeling that included the following: (a) greatly enhanced influx of calcium via the L-type calcium channels; (b) the marked suppression or absence of I_Ca_- and caffeine-triggered Ca^2+^ release; (c) the absence of SR calcium release signals that generally accompany spontaneously activated cytosolic calcium transients in WT cells; and (d) calcium transients that were slowly developed and activated by high caffeine concentrations (20 mM) that were variably suppressed by IP3 or hemi-channel blockers or by an inhibitor of mitochondria. These findings suggested the remodeling of the calcium signaling pathway to one dominated by an influx of calcium through the transmembrane channels and/or Ca^2+^ release from other cellular Ca^2+^ pools. To further explore the mechanism of CICR suppression and determine whether calcium signaling remodeling also occurs in animal models carrying this mutation, we attempted to generate a knock-in mouse model expressing the RyR2-Q3924E mutation that corresponds to the human Q3925E mutation. Little has been reported on the pathophysiology of the human Q3925E mutation, except for one case of a 9-year-old girl who suffered from arrhythmia and experienced sudden death [18,19], or its calcium signaling consequences in animal models. Mice carrying RyR2-Q3924E+/− did not survive more than 3 weeks, preventing the establishment of a germline colony. Animals carrying up to a 25% mutation died of severe cardiac hypertrophy and arrhythmia.

## 2. Materials and Methods

### 2.1. Maintenance and Differentiation of hiPSCs

Human-induced pluripotent stem cells were maintained in Stemflex medium (Gibco) and passaged routinely with accutase on vitronectin (Gibco)-coated plates at 37 °C with 5% (vol/vol) CO_2_. hiPSCs were differentiated into cardiomyocytes as described previously [17]. Briefly, dissociated hiPSCs were plated in 24-well plates and then treated with 6 μM CHIR99021 (Tocris), a GSK3β inhibitor, for 24 h in RPMI/B-27 without insulin. Seventy-two hours after CHIR99021 treatment, 5 μM IWR1, a Wnt signaling inhibitor, was added to the same culture media for 48 h. Cells were then maintained in RPMI/B-27 culture media without insulin and switched to RPMI/B-27 medium containing insulin (B27(+) medium) when majority of the cells started beating. Spontaneously contracting clusters were then enzymatically (TrypLE Select Enzyme [10×], Gibco) dissociated into single cardiomyocytes for electrophysiological and Ca^2+^ imaging experiments.

### 2.2. Generation of RyR2-Q3924E Mouse Model

To model a *RyR2*-p.Gln3924Glu polymorphism in the highly conserved RyR2 Ca^2+^ binding domain that is reported to be associated with sudden cardiac death, we used CRISPR-mediated genome editing in mouse embryos to generate a *Ryr2*-p.Gln3924Glu (Q3924E) substitution in the homologous region of mouse RyR2 (exon 87 of RyR2). Synthetic guide RNA (sgRNA; CACGCTCACGGAGTACATCC) and single-stranded oligonucleotide (ssODN; 5′ CTTTTCC AAAGCCATTC AAGTGGCGAA GCAG GTGTTCAACACGCTCACGGAaTAtATCgAGGTAAGCAGCAAGCATTTCAGTTGTGTAAAGAGAGACAGAGACAACCGTTTTTAAAG) were designed and synthesized by MUSC Transgenic & Genome Editing (TGE) Core facility. CRISPR reagents were delivered by electroporation (EP) to single-cell embryos using 5 mm cuvettes and an NEPA21 Super Electroporator (Nepa Gene Co., Chiba, Japan). Electroporation conditions were as follows: the poring pulse was set to 200 V, 1 ms pulse width, 50 ms pulse interval and +4 pulse number. The transfer pulse was set to 20 V, 50 ms pulse width, 50 ms pulse interval and ±5 pulse number (attenuation rate was set to 40%). One day later, the electroporated embryos that had developed to the 2-cell stage were transferred into the oviducts of pseudopregnant female ICR mice, and newborns were obtained. Genomic DNA sequences around Exon 87 of the *Ryr2* gene were determined using the following primers: 5′-GTAGGGGTGTTGCTGGAGAT-3′ and 5′-AACTTGGCCATGTTGCTGAG-3′.

### 2.3. Hematoxylin and Eosin Staining

The hearts from the RyR2-Q3924E mutant mice and their WT littermates were fixed with 4% paraformaldehyde solution in PBS. The fixed hearts were then dehydrated through graded alcohol changes from 70% to 100%. Hearts were cleared in toluene and embedded in paraffin. Hearts were then cut across their long axis to show the four chambers or short axis with a 5 μm thickness using the Leica microtome and then mounted onto slides. Slides were then deparaffinized in xylenes and rehydrated through graded alcohols and distilled water before hematoxylin and eosin staining. Images were acquired using Keyence microscope and TIRF microscope.

### 2.4. RNA Sequencing

Total RNA was extracted from WT and mutant hiPSC-CMs using TRIzol Reagent. RNA-Seq was performed using Novogene’s Illumina NovaSeq PE150 for next-generation sequencing according to the manufacturer’s directions. Sequenced reads in Fastq format were aligned to hg38 reference genome using STAR (Spliced Transcripts Alignment to a Reference) alignment software (https://www.encodeproject.org/software/star/). R package DESeq2 was used to normalize and quantify the aligned RNA-seq reads with threshold *p* ≤ 0.05 and log2 fold change ≤ −0.5 to compare the treatment groups to each other. Heatmaps were created using R package Pheatmap version 1.0.12. 

### 2.5. Quantitative RT-PCR

Total RNAs of WT and mutant hiPSC-CMs were extracted with TRIzol LS reagent (Ambion, Life Technologies, Carlsbad, CA, USA) and chloroform. The cDNAs were synthesized from total RNAs by reverse transcription with Verso cDNA Synthesis Kit (Thermo Fisher Scientific, Waltham, MA, USA). Quantitative PCR was performed using Applied Biosystems™ SYBR™ Green PCR Master Mix (Applied Biosystems, Waltham, MA, USA) in the CFX Connext Real-Time PCR Detection System (Bio-Rad, Hercules, CA, USA). The protocol for qPCR was 95 °C for 10 min, followed by 40 cycles of 95 °C for 15 s and 60 °C for 1 min. To obtain the relative quantization, the ADRB1 mRNA expressions were normalized to those obtained for the corresponding GAPDH expression levels. Each sample was run with three replicates.

Primers for GAPDH:

Forward: 5′-GTCTCCTCTGACTTCAACAGCG-3′;

Reverse: 5′-ACCACCCTGTTGCTGTAGCCAA-3.

Primers for ADRB1:

Forward: 5′-TTCCTGCCCATCCTCATGCACT-3′;

Reverse: 5′-GTAGAAGGAGACTACGGACGAG-3′.

### 2.6. R-CEPIA1er and ER-GCaMP6-150 Ca^2+^ Probes

pCMV R-CEPIA1er plasmid (#58216) was purchased from Addgene (Watertown, MA, USA) [20]. We used ER-GCaMP6-150 Ca^2+^ probe as described previously [17]. Adenovirus carrying R-CEPIA1er was produced by Welgen Inc. (Worcester, MA, USA). Dissociated hiPSC-CMs were infected by a medium containing R-CEPIA1er adenovirus containing an MOI of 300–500 virus particles per cell (vp/cell). After 6–8 h, the virus medium was removed, and cells were supplemented with B27+ medium and kept in the incubator at 37 °C and 5% CO_2_ until they were used between 48 and 72 h post-infection.

### 2.7. Statistical Analysis

Results are indicated as the means ± SEM. Comparative analysis was determined using one-way analysis of variance (ANOVA) followed by Tukey’s test or Student’s *t*-test.

Significant differences are labeled with one (*p* < 0.05, *) or two stars (*p* < 0.01, **).

Origin 8 was used to obtain statistical values.

## 3. Results

### 3.1. Human Stem-Cell-Derived Cardiomyocytes Harboring the Q3925E Mutation

In a previous article [17], we reported that hiPSC-derived Q3925E mutant myocytes had the following: (1) strongly suppressed SR calcium release signals; (2) no significant differences in the magnitude of spontaneous cytosolic Ca^2+^ transients, even when expressing irregular arrhythmic auto-pacing; (3) the absence of or greatly suppressed calcium release triggered by 5 mM caffeine; (4) unchanged or somewhat smaller 4-CMC-induced calcium release signals; and (5) slowly activating responses triggered by 20 mM caffeine that were variably suppressed in different cells by IP3, hemi-channel or mitochondrial inhibitors. These properties are schematically illustrated in Figure 1 comparing the calcium signaling phenotype of Q3925E mutant cells to wild-type hiPSC-CMs. Note that while the 4-CmC and caffeine triggered equivalent calcium transients in WT myocytes, calcium transients activated by either drug were significantly suppressed in mutant cells, as shown in Figure 1B,C. Spontaneously generated calcium transients recorded simultaneously from the cytosol and SR or activated by the rapid application of 5 mM caffeine or 4-CmC (measured simultaneously with Fluo-4 and a genetically engineered and targeted ER/SR R-CEPIA1er probe) produced robust SR signals in WT but not in mutant cells, as shown in Figure 1D. The data suggested that mutation in the RyR2 calcium binding site suppresses CICR activated by an influx of calcium on L-type calcium channels or triggered by caffeine. To check whether the RyR2 Ca^2+^ binding site mutations had altered the SR calcium leak, we measured the diastolic Ca^2+^ leak by exposing the myocytes to 1 mM tetracaine and zero Ca^2+^ zero Na^+^, followed by rapid exposures to 4-CmC in ER-GCaMP6-infected and Fura-2-incubated spontaneously beating cells. Figure 1E shows that tetracaine rapidly suppressed the ER Ca^2+^ signal generated by release of Ca^2+^ from SR, resulting in a decrease in cytosolic calcium monitored by Fura-2 in WT cells. In sharp contrast, tetracaine failed to have a significant effect on the SR leak signal in mutant cells, a finding consistent with the absence of spontaneously igniting calcium sparks in Q3925E hiPSC-CMs, reported by us earlier in the mutant cells [17], suggesting suppressed ER Ca^2+^ release and smaller SR Ca^2+^ stores.

### 3.2. RNA Sequencing and Quantitative RT-PCR

To test if the calcium signaling genes of Q3925E mutant cells were altered, we measured RNAseq heatmaps of calcium ion homeostasis genes which included the following: the calcium ion transport genes, the positive regulation of cardiac contraction genes and regulation of cellular response to stress genes. Figure 2A,B show that the majority of RNA-seq genes had a decreased expression of the *calcium homeostasis gene (GO 0055074) and the calcium ion transport genes (GO 0006816)* in Q3925E mutant hiPSC-CMs as compared to WT cardiomyocytes. We also found the downregulation of *positive regulation of heart contraction genes (GO 0045823)*, including *adrenergic receptor B1(ADRB1), APLN and SMTN genes*, as shown in Figure 2C, suggesting suppressed β-adrenergic inotropic responsiveness and smooth muscle cell contraction in Q3925E mutant hiPSC-CMs, consistent with totally suppressed CICR. In sharp contrast, the *regulation of cellular response to stress genes (GO 0080135)* increased, as shown in Figure 2D.

Quantitative RT-PCR analyses showed that the expression levels of the ADRB1 messenger RNA in E3848A mutant hiPSC-CMs were significantly lower compared to those of the WT cells. The Q3925E cells’ ADRB1 expression was also lower compared to WT cells but not statistically significant, as shown in Figure 2C. Note, however, that the cycle threshold values for ADRB1 are high, with 25–30 for WT, 32–38 for E3848A, 29–39 for Q3925E and 14.7–18.6 for GAPDH, the housekeeping gene of all three groups, suggesting the relatively poor expression of ADRB1. Our findings are consistent with reports that adrenergic stimulation and exercise stress testing often fail in RyR2 loss of function and calcium release deficiency syndrome [21,22,23].

### 3.3. Generation of Q3924E Mice

Since Q3925E-RyR2 mutation causes marked Ca^2+^ signaling aberrancies in hiPSC-CMs, suggesting the possible remodeling of calcium signaling pathways, we sought to create a mouse model of this mutation to further explore its whole animal pathology. Using the CRISPR/Cas9 technique, we introduced the RyR2-Q3924E mutation (equivalent to Q3925E in humans) in mouse embryos with the technical assistance of the MUSC Transgenic Core facility. Figure 3A shows the design to introduce the Q3924E mutation (CAG to GAG) and two silent mutations (TAC to TAT and GAG to GAA) in *RYR2* exon 87. Genomic DNA was collected from the mice toe tissue, and the gene-edited locus was amplified by PCR. The mutations were screened by the mutant primers (Figure 3A) and then confirmed by the sequencing of the WT PCR product, in which the primer binding sites were outside of the homologous arms. We obtained three male heterozygous mice and one male chimera (mosaic) mouse harboring the Q3924E mutation. Figure 3 shows the sequencing results of the four mice. Heterozygous founders B1, E1 and F3 all showed the desired mutation in one of their alleles, carrying 50% of the Q3924E mutation. The mosaic mouse founder A9 carried only 25% of the Q3824E mutation. Surprisingly, all the heterozygous and chimera mice died before we could establish the germline transmission of the mutation in their offspring. A postmortem examination of the hearts, nevertheless, showed that the hearts of the chimera mice were significantly larger than those of the wild-type mice (160 mg vs. 105 mg).

### 3.4. Histopathology of Q3924E Mouse Heart

The heart of the A9 founder mouse was fixed and longitudinally sectioned. Cross-section samples were obtained from the E1 founder mouse heart. H&E-stained longitudinal heart sections from the WT and founder A9 chimera mice showed significant cardiac hypertrophy in the Q3924E mutants compared to the WT mice, as shown in Figure 4A. Figure 4B also compares WT vs. founder F3 (heterozygous) mouse heart cross-sections, showing a thicker left ventricular wall and a much smaller ventricular chamber in the Q3924E mice. The imaged cardiomyocytes also appeared to be larger and somewhat disorganized, as shown in Figure 4C. Although we have thus far examined only a few hearts, the data obtained suggest that the Q3924E-RyR2 mutation causes significant cardiomyopathy, which is lethal to mice.

## 4. Discussion

### 4.1. RyR2 Calcium Binding Site Mutations

Point mutations in RyR2 protein have been extensively studied in both hiPSC-CMs and HEK293 cells [10,11,16,17,24]. We have already reported on the EC coupling consequences of mutations in the RyR2 calcium binding site expressed in hiPSC-CMs [17]. As expected, mutating any one of the five residues in the calcium binding site (Q3925E or E3848A) caused a major disruption in calcium signaling that resulted in the suppression of CICR and the loss of caffeine-triggered calcium release. What was most unexpected was that mutant cardiomyocytes continued beating spontaneously, though arrhythmically, and responded to high caffeine concentrations by slowly releasing calcium, which was variably sensitive to IP3, hemi-channels and mitochondrial inhibitors in different myocytes, suggesting the possible epigenetic remodeling of CICR that allows cardiomyocytes to maintain their spontaneous contractile function. This finding, coupled with one case report of a nine-year-old girl expressing the Q3925E lethal mutation [18,19], motivated us to create a mouse model of the Q3924E mutation to better understand the in vivo pathology of the whole heart.

To our knowledge, this is the first reported attempt at creating transgenic mice carrying the RyR2 Ca^2+^-binding site mutation Q3924E (equivalent to Q3925E in humans) which causes sudden death. We used CRISPR/Cas9 gene editing to introduce point mutations in mouse embryos that had developed to the two-cell stage. Unfortunately, all the founder heterozygous mice died unexpectedly. We could only carry out postmortem histological studies that included fixation and staining of the hearts from dead founder animals. The histological examination showed the development of cardiac hypertrophy in the Q3924E mice, consistent with the loss-of-function mutation and sudden cardiac death of only one known case of a 9-year-old girl expressing this mutation who experienced unexplained sudden death.

### 4.2. Comparison of hiPSC-CMs and Animal Model

In CRISPR/Cas9-gene-edited RyR2-Q3925E hiPSC-CMs, we found significantly suppressed I_Ca_-triggered Ca^2+^ release and caffeine-triggered calcium release. Spontaneous beating, mostly arrhythmic, persisted in mutant cells without the SR Ca^2+^-release signals, suggesting other dormant calcium signaling pathways of cardiomyocytes are activated, allowing the mutant myocytes to survive and beat spontaneously. RNA-seq of Q3925E hiPSC-CMs suggests the downregulation of positive regulatory cardiac contraction genes, calcium ion homeostasis and transport genes. Since the Q3924E mutation induced sudden cardiac death, we were unable to dissociate the cardiomyocytes and characterize the Ca^2+^ signaling properties of the myocytes of the founder mouse hearts. H&E staining of the Q3924E mutant hearts, however, showed significant cardiac hypertrophy of the left ventricle and a thicker left ventricular wall. Our attempts to determine whether similar EC coupling remodeling was also occurring in the mouse hearts were unsuccessful because all the heterozygous and mosaic mice died within 3 weeks of life, suggesting that mutant caridomyocytes could not support adequate cardiac function to support life.

### 4.3. Strength and Weaknesses of Arrhythmia Models: hiPSC-CM vs. Transgenic Mouse vs. Recombinant RyR2 HEK 293 Cell Line Models

Creating CRISPR/Cas9-gene-edited mutations in control hiPSC-CMs or generating hiPSC-CMs directly from the fibroblasts of patients that express the mutation [16,17,24], isolated cells from mutant mice hearts carrying the mutation [8,9,25,26] or HEK293 cell lines transfected with recombinant mutant RyR2 proteins [10,11,27] are the platforms generally used for studying CPVT1 arrhythmia.

The recombinant platform addresses the functional aberrancies caused by a particular mutation on cell function and, as such, provides the most direct approach to studying the structure/function effects of the RyR2 protein mutation on cellular level. Nevertheless, the use of HEK293 epithelial kidney cell lines is limited with respect to calcium signaling pathways even after transfecting them with recombinant mutant RyR2 protein because HEK293 cells do not express the multiple cardiac Ca^2+^ signaling pathways that exist in mammalian hearts. The recombinant HEK cell approach may therefore be inadequate in expressing the epigenetic remodeling of cardiac signaling pathways that would occur in response to a mutation of proteins critical for the CICR signaling of myocytes. It should be noted that even though CICR is the dominant calcium signaling pathway in mammalian cardiomyocytes, the cells also express other calcium signaling pathways that include ER/IP3, mitochondrial and plasmalemmal calcium transporting proteins that may be activated when CICR becomes dysfunctional.

The mouse model, on the other hand, while addressing the arrhythmogenic consequences of mutation in vivo, suffers from both heart-size differences and the variable expression of ion channels between human and mouse hearts. The differential ion channel expression and length of conduction pathways are two critical aspects in triggering arrhythmia in the human heart. There are also great difficulties with longevity due to generating transgenic mouse lines of mutations that cause heart failure and sudden death.

In sharp contrast to the mouse model, the hiPSC-derived cardiomyocyte platform presents a model with a human genetic background that expresses the ion channel composition and Ca^2+^ signaling profiles of adult human cardiomyocytes and, as such, is a reliable model to study human pathology, were it not for the immaturity of the created cells. The structural immaturity of the cells is reflected in the absence of a t-tubular system, the flat cellular shapes of myocytes and large cell surface to volume ratios, which are likely to contribute to their spontaneous beating. Although the spontaneous beating of hiPSC-CMs argues for their immaturity, similar to embryonic and neonatal cardiomyocytes, it should be noted that adult ventricular myocytes, when kept in culture media longer than a few days, also develop spontaneous beating as they develop flat shapes with pseudopods. Note also that hiPSC-CMs do not express significant levels of HCN2/HCN4 genes encoding the funny current(*If*), in sharp contrast to SA-nodal cells, which is consistent with the possibility that other pacing mechanisms may be involved in the spontaneous beating of hiPSC-CMs.

Thus, the three platforms used as models of human arrhythmias provide different vistas for studies of arrhythmias, suggesting that a combined approach using hiPSC-CMs and transgenic mice might best approximate human CPVT1 arrhythmias or Ca^2+^ release deficiency syndrome.

## 5. Conclusions

The human RyR2-Q3925E mutation has been reported to be associated with unexplained sudden death. Mutant hiPSC-CMs carrying the Q3925E mutation showed suppressed CICR and the loss of caffeine-triggered calcium release but the persistence of spontaneous beating, suggesting the remodeling of EC coupling. The transgenic mice carrying the Q3924E-RyR2 mutation died within 3 weeks, preventing the establishment of germline animals. Postmortem H&E staining of Q3924E mutant hearts showed significant myocyte disorganization and massive cardiac hypertrophy. Thus, human Q3925E/mouse Q392E mutations are lethal, causing sudden death at an early age, which is consistent with one reported human case.

## Figures and Tables

**Figure 1 cells-13-02051-f001:**
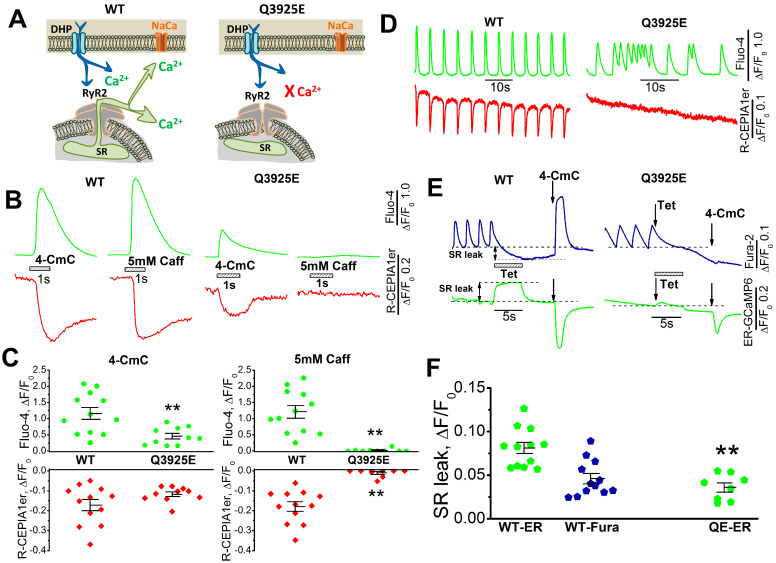
Calcium signaling from WT and Q3925E hiPSC-CMs. (**A**) The diagram shows that RyR2 Ca^2+^-binding residues mutation cause defective CICR. (**B**) Application of 5 mM 4-CmC and 5 mM caffeine triggered ER Ca^2+^ release and cytosolic Ca^2+^ signal measured by R-CEPIA1er and Fluo-4 simultaneously. (**C**) Quantification of 4-CmC and caffeine-triggered Ca^2+^ signals measured by Fluo-4 and R-CEPIA1er simultaneously. n = 12 for WT and n = 10 for Q3925E cells. ** *p* < 0.01 vs. WT by unpaired *t*-test. (**D**) Spontaneous cytosolic Ca^2+^ transients and SR Ca^2+^ release measured by Fluo-4 and R-CEPIA1er in WT and Q3925E cells. (**E**) SR Ca^2+^ leak measured with Fura-2 and ER-GCamP6 probe in WT and Q3925E mutant cells. (**F**) Quantification of SR Ca^2+^ leak measured by ER-GCamMP6 and Fura-2 in WT cells and SR Ca^2+^ leak measured by ER-GCamMP6 in Q3925E cells. ** *p* < 0.01 vs. WT by unpaired *t*-test.

**Figure 2 cells-13-02051-f002:**
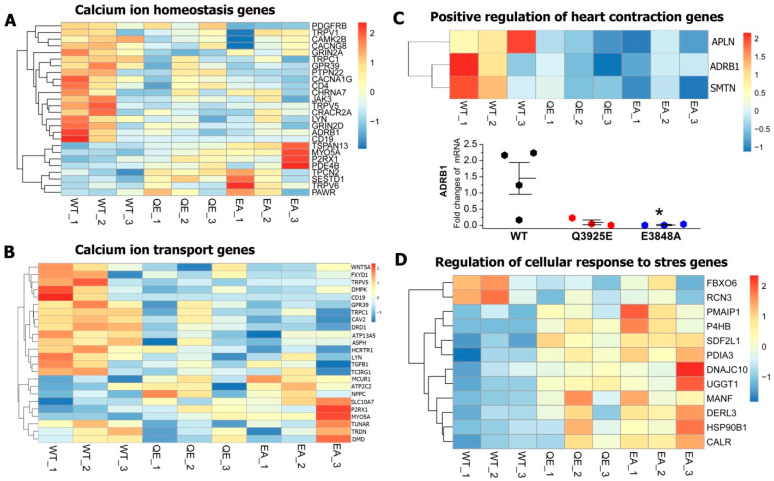
Heatmap of RNA sequencing results from WT and Ca^2+^ binding site mutant Q3925E and E3848A hiPSC-CMs. (**A**) GO: 0055074. Calcium homeostasis genes. (**B**) GO:0006816. Calcium ion transport genes. (**C**) GO: 0045823. Positive regulation of heart contraction genes. Lower panel: quantification of ADRB1 transcription levels in WT and the two mutant hiPSC-CMs by quantitative RT-PCR. * *p* < 0.05 vs. WT by one-way ANOVA. (**D**) GO: 0080135. Regulation of cellular response to stress genes. The lines on the left of the heatmaps are a simple dendrogram, which attempts to find trends within the data and cluster them together.

**Figure 3 cells-13-02051-f003:**
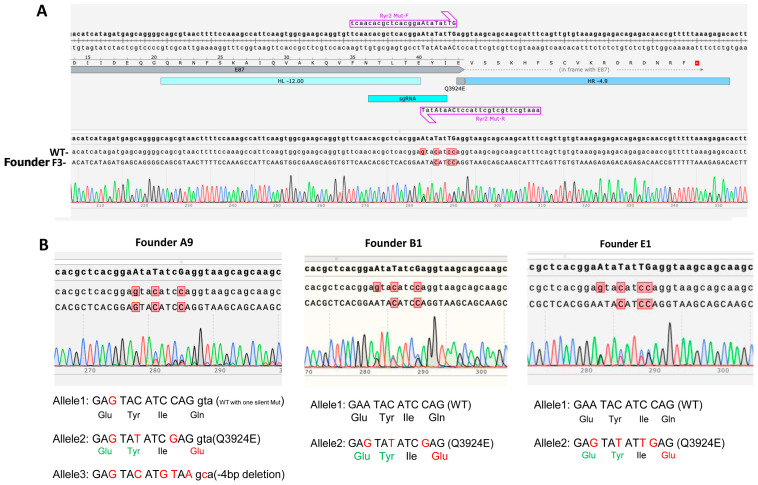
Generation of mutant mice harboring RyR2-Q3924E. (**A**) CRISPR/Cas9 design and genome sequences of heterozygous founder F3. CRISPR/Cas9 introduced 4 mutations, 5′-GA**A**TA**T**AT**TG**AG-3′, in mouse Ryr2 gene. One results in Q3924E mutation (CAG to GAG), and the other 3 are silent mutations (no amino acid changes). Two different peaks were observed at four nucleotide positions in Sanger sequencing, because the heterozygous mouse carry both wild-type and mutant sequences. (**B**) Genome sequencing results of founders A9, B1 and E1. For founders A9 and B1, 3 mutations, 5′-GA**A**TA**T**ATC**G**AG-3′, were introduced in mouse Ryr2 gene. One results in Q3924E mutation (CAG to GAG), and the other two are silent mutations. Three different peaks are observed in A9 because the chimera mouse carried 25% of the Q3924E mutation. Founders B1 and E1 are both heterozygous.

**Figure 4 cells-13-02051-f004:**
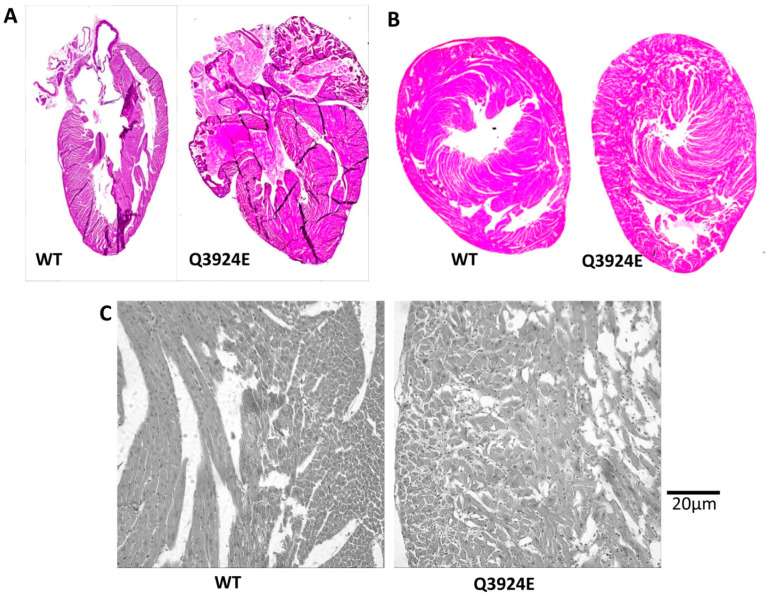
Histological analysis of heart sections from RyR2-WT and RyR2-Q3924E+/− mutant mice. (**A**) H&E-stained longitudinal direction heart sections of WT and mutant hearts (founder A9). Images were captured from Keyence microscope. (**B**) Cross-section of WT and mutant heart (founder F3). (**C**) Comparison of cardiomyocytes from the same area of left ventricle of WT and mutant heart sections from panel (**B**). Images were captured from TIRF microscope with 20× objective lens.

## Data Availability

The data underlying this research will be shared upon reasonable request to the corresponding author.

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
