# Peer review of "Attempts to Create Transgenic Mice Carrying the Q3924E Mutation in RyR2 Ca^2+^ Binding Site"

_cells, 2024, doi:10.3390/cells13242051_

Round 1

Reviewer 1 Report

Comments and Suggestions for Authors

The current study is a detailed analysis of the consequence of a previously identified mutation of the RyR2 in cardiomyoctyes. The study continues a previous study and the main important advantage of this study is the transfer to genetic mice models.

Major comments:

As Fig. 1 is a repetition of previously published data (please indicate this in the Figure legend) the new data start from Fig. 2. Here heat maps are presented from RNA Seq. data. Although the main points may be taken from this part of the study the data presented here have many disadvantages as well. First, only two samples per condition (WT, OW, And EA) are analyzed. Second, the RNA Seq quantification is a little bit semi-quantitative and main conclusions require confirmation via an independent method (preferentially qRT-PCR). Third, it is unclear what the level of expression really means. What is -1.5? I would expect a 1.5-fold lower expression. But to what have these data to be compared? Why are the WT controls not normalized to zero and what does zero mean? Fourth, ADRB1, one of the key findings, is presented twice: In Fig. 2B it is part of calcium ion transport genes, what is properly wrong for a G-protein coupled receptor. In Fig. 2C it is part of genes with positive regulation of heart contraction. However, although the same gene is plotted twice the data of the same samples differ significantly. How do you explain this? Fifth, the lines on the left side of the figure are not explained in the figure legend. Overall, the data are important for the study but not really to understand and based on an unacceptable low level of samples.

The preliminary data about the three mouse strains are very interesting. I do not understand why the authors did not validate their expression data from the iPS cells by RT-PCR in these tissues, i.e. the expression of ADRB1. The authors must validate their findings from Fig. 2 with these tissues and this would make the study much stronger.   

Reviewer 2 Report

Comments and Suggestions for Authors

MANUSCRIPT: 3218316

TITLE: Attempts to create transgenic mice carrying the Q3925E mutation in RyR2 Ca2+ binding site

 The manuscript 3218316 “Attempts to create transgenic mice carrying the Q3925E mutation in RyR2 Ca2+ binding site” presents a very well organized work in which the main issue of this research is to analyze and evaluate how the mutation in RyR2 Ca2+ binding site Q3925E in rats can give rise significant cardiac hypertrophy and relate the suggesting Q3924E-RyR2 mutation that was lethal in the mouse with the reported evidence that human RyR2-Q3925E mutation has been associated with unexplained death.

The topic of this work is relevant and adds a new significant information in this subject area of ​​knowledge because the proposed methodology allows identifying the mutation in RyR2 Ca2+ binding site Q3925E and relating it to the histopathological analysis of the heart which shows that death can occur be related to cardiac hypertrophy resulting from the Q3924E-RyR2 mutation.

In my opinion:

This work is well structured, well planned and the research is competently carried out.

 Methodology used was adequate and is suitable for the work carried out.

Results and discussion are properly discussed.

Conclusions are in accordance with the objectives of the investigation and the results obtained.

References are appropriate and also updated with around 30 % of references data from the last 5 years.

Regarding the manuscript, I have no additional comments or suggestions for the authors.

Author Response

Response to reviewer #2.

We thank the reviewer for careful reading of the MS and his positive remarks.

Reviewer 3 Report

Comments and Suggestions for Authors

The article titled "Attempts to create transgenic mice carrying the Q3925E mutation in RyR2 Ca2+ binding site" presents novel insights into the role of the Q3925E mutation in cardiac arrhythmia, building on previous work with human cardiomyocytes. The use of CRISPR/Cas9 to create a knock-in mouse model is a commendable methodological approach, reinforcing the study's relevance in advancing our understanding of the RyR2 mutation's in-vivo effects. However, the text could benefit from minor revisions for clarity. Overall, the study is a valuable contribution to the field, despite challenges in germline transmission.

1. I understand that Q3924E mutation is the murine equivalent of Q3925E mutation in human cells. However, considering the notable similarity between the two designations, they could easily be mistaken in certain sections of the manuscript. Importantly, the title of the manuscript could appear misleading considering that transgenic mice generated in the work carried Q3924E mutation (and not Q3925E mutation as indicated in the title).

2. The entire text could benefit from technical editing, considering the presence of extra spaces throughout. There are some inconsistencies in font size in Materials and Methods section. Certain phrases have been underlined, which delineate from the appropriate formatting of the text.

3. Figure 1 could have been presented in a larger size. I had difficulty reading the labels in the figure.

Author Response

  1. I understand that Q3924E mutation is the murine equivalent of Q3925E mutation in human cells. However, considering the notable similarity between the two designations, they could easily be mistaken in certain sections of the manuscript. Importantly, the title of the manuscript could appear misleading considering that transgenic mice generated in the work carried Q3924E mutation (and not Q3925E mutation as indicated in the title).

Response: We Thank the reviewer for pointing this out and have modified the title according to the reviewer's suggestion. 

  1. The entire text could benefit from technical editing, considering the presence of extra spaces throughout. There are some inconsistencies in font size in Materials and Methods section. Certain phrases have been underlined, which delineate from the appropriate formatting of the text.
  2. Figure 1 could have been presented in a larger size. I had difficulty reading the labels in the figure.

 Response: We have carried out a technical editing of the MS and changed figure 1 size as per the reviewer’s suggestion. We thank the reviewer for helping to improve the MS.

Round 2

Reviewer 1 Report

Comments and Suggestions for Authors

I thank the authors for clarification of my points. Nevertheless with only having these data the overall conclusion is tto strong in my opinion.

Author Response

We have indicated the preliminary nature of our conclusions and some of our data, by titling our MS as “An attempt to create” Attempts to create transgenic mice carrying the Q3924E mutation in RyR2 Ca2+ binding site.